# Enhancing Animal Production through Smart Agriculture: Possibilities, Hurdles, Resolutions, and Advantages

**Moammar Dayoub** [1] , **Saida Shnaigat** [2], **Radi A. Tarawneh** [3], **Azzam N. Al-Yacoub** [4], **Faisal Al-Barakeh** [5] and **Khaled Al-Najjar** [6,*]

1   Faculty of Technology, University of Turku (UTU), 20500 Turku, Finland; moammar.dayoub@utu.fi
2   Department of Nutrition, Faculty of Agriculture, Jerash University, Jerash 26150, Jordan; saidashnaigat@yahoo.com
3   Faculty of Agriculture, Jerash University, Jerash 26150, Jordan; radi.amtarawneh@yahoo.com
4   Department of Biology, Faculty of Applied Science, Umm Al-Qura University, Makkah 24382, Saudi Arabia; anyacoub@uqu.edu.sa
5   Livestock Research Directorate, National Agriculture Research Center, Amman 11118, Jordan; falbarakeh@yahoo.com
6   General Commission for Scientific Agricultural Research, GCSAR, Damascus 12573, Syria
*   Correspondence: khnajj2011@yahoo.com

**Abstract:** Smart livestock farming utilizes technology to enhance production and meet food demand sustainably. This study employs surveys and case studies to gather data and information, subsequently analyzing it to identify opportunities and challenges. The proposed solutions encompass remote sensing, technology integration, farmer education, and stakeholder engagement. The research delves into smart technologies in animal production, addressing opportunities, challenges, and potential solutions. Smart agriculture employs modern technology to improve efficiency, sustainability, and animal welfare in livestock farming. This includes remote monitoring, GPS-based animal care, robotic milking, smart health collars, predictive disease control, and other innovations. Despite the great promise of smart animal production, there are existing challenges such as cost, data management, and connectivity. To overcome these challenges, potential solutions involve remote sensing, technology integration, and farmer education. Smart agriculture provides opportunities for increased efficiency, improved animal welfare, and enhanced environmental conservation. A well-planned approach is crucial to maximize the benefits of smart livestock production while ensuring its long-term sustainability. This study confirms the growing adoption of smart agriculture in livestock production, with the potential to support the sustainable development goals and deliver benefits such as increased productivity and resource efficiency. To fully realize these benefits and ensure the sustainability of livestock farming, addressing cost and education challenges is essential. Therefore, this study recommends promoting a positive outlook among livestock stakeholders and embracing smart agriculture to enhance farm performance.

**Keywords:** smart livestock farming; technology integration; sustainable development; stakeholder engagement

## 1. Introduction

Smart agriculture employs technology to enhance livestock productivity and efficiency. It achieves this by enhancing animal health, productivity, feed quality, traceability, and sustainability in meat and dairy production. Additionally, smart agriculture reduces the consumption of resources, such as water and land, while enhancing environmental quality. As a result, smart agriculture effectively enhances livestock management, leading to higher productivity and efficiency in livestock production, improved animal product quality, and increased environmental sustainability [1]. Smart agriculture, through technological advancements, elevates livestock management, resulting in increased productivity, superior

product quality, and improved environmental sustainability. It signifies a transformative shift in the livestock industry towards a more efficient and sustainable future.

Smart agriculture blends cutting-edge technology with traditional farming to increase efficiency and productivity in livestock production. It uses remote sensing, artificial intelligence, and data analysis to optimize farm management and maximize production and quality. Smart agriculture employs technologies such as sensors, robots, drones, and artificial intelligence to achieve these goals [2–4]. Smart agriculture integrates technology with traditional farming to optimize livestock production by using sensors, robots, drones, and AI for improved efficiency and product quality.

Specific applications include monitoring animal health, automating tasks like feeding and watering, and collecting data on health and feed consumption and analyzing it to make informed production and management decisions. Smart agriculture greatly boosts farm efficiency, productivity, and sustainability, ensuring a secure and nutritious supply of animal products for the future [5–7]. Smart agriculture apps offer animal health monitoring, automated feeding, and data collection, revolutionizing farming for improved efficiency and sustainability.

Smart agriculture brings multiple benefits to livestock farms, including improved animal health and welfare, increased productivity, and reduced environmental impact. Technological advancements ensure that smart agriculture will be crucial in addressing livestock production challenges and enhancing production quality and efficiency [8–10]. Smart agriculture is a promising technology with the potential to improve the livestock industry significantly.

Smart agriculture transforms livestock farms, enhancing animal welfare, productivity, and sustainability. Ongoing technological advancements ensure it remains a vital solution for production challenges, paving the way for a more efficient and eco-friendly future in the livestock industry.

Applying smart agriculture to improve animal production is crucial, as it harnesses cutting-edge technologies and data-driven methods to revolutionize animal farming. This topic holds significance in advancing animal welfare, resource efficiency, and agricultural sustainability. It provides potential solutions to meet the increasing global demand for food while mitigating environmental impact. Furthermore, it offers opportunities for farmers and industry stakeholders to elevate their practices and adapt to the ever-changing landscape of modern agriculture [11–13]. Smart agriculture is the future of animal farming, with the potential to transform it into a more sustainable, efficient, humane, and productive industry. It harnesses cutting-edge technologies and data-driven methods to make animal farming more precise and optimized while reducing its environmental impact.

Smart animal farming enhances welfare, efficiency, and sustainability through technology and data, addressing global food demand and environmental concerns with innovative solutions for a brighter agricultural future.

The problem statement states that unequal access to and affordability of smart farming tools in livestock production is a barrier to achieving sustainable food production and animal welfare.

Smart agriculture holds great promise for livestock farming by enhancing efficiency, sustainability, and animal welfare. Addressing cost, data management, and connectivity challenges is crucial for its widespread adoption. Implementing the suggested solutions can help the agricultural sector unlock the full potential of smart livestock farming, contributing to the sustainable development goals and improving overall farm performance [14].

*Research Questions*

Question 1: Explore the applications of smart agriculture in livestock farming.

Question 2: Delve into the practical applications of smart animal production technologies that enhance animal welfare, improve productivity, and promote sustainability.

Question 3: Unravel the potential of Blockchain technology to enhance efficiency and transparency in livestock production.

Question 4: Examine the opportunities, challenges, solutions, and benefits associated with implementing smart livestock production farms.

Question 5: Investigate how smart agriculture technologies can contribute to sustainable livestock production practices.

This study aims to improve animal farming for both the environment and farmers. This benefits both farmers and the broader agriculture industry; it entails a comprehensive examination of smart technologies in animal production, including an exploration of their opportunities and challenges. This study offers directions for future research and development in this field.

## 2. Materials and Methods

This research delves into cutting-edge animal production technologies, emphasizing opportunities, challenges, solutions, and benefits, particularly in the realm of smart agriculture applied to livestock. By scouring databases and peer-reviewed journals, the researchers pinpointed studies on smart livestock farming using search-specific keywords. They assessed these studies based on their design, methodology, results, and relevance, extracting essential findings and concepts from the pool of previous research.

### Review of Previous Studies on Smart Animal Production

The researchers conducted a thorough literature review using a systematic approach, incorporating a range of academic sources. They gathered data from research institution websites, government agencies, and journals. The data extraction process was meticulous, paying close attention to the collection of source information. The analysis prioritized findings supported by strong and consistent evidence from prior studies. Out of the 235 research papers collected, 33 were excluded after scrutinizing their summaries due to their findings not aligning with the current research objectives. This selective process was crucial to maintain a focus on relevant evidence and outcomes.

Smart livestock production leverages technology to enhance animal health, nutrition, monitoring, management, and marketing, resulting in various economic and social benefits (Figure 1).

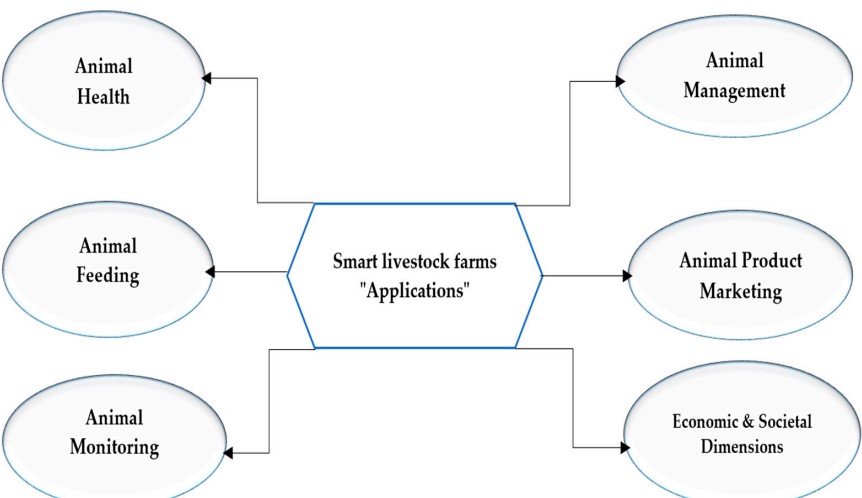

**Figure 1.** Flowchart of the primary smart agricultural applications in livestock production.

## 3. Results and Discussion

### 3.1. Smart Agriculture in Animal Production

Animal healthcare and safety are undergoing a transformative evolution, leveraging technology to enhance the well-being of animals. The integration of smart devices and artificial intelligence contributes significantly to monitoring and improving animal health

by identifying patterns and trends. The benefits of smart animal healthcare and safety are evident in early disease detection [15,16].

Biotechnology plays a pivotal role in advancing animal health, with its positive impact extending to the development of new vaccines, drugs, and effective parasite control methods. The integration of smart systems further enhances animal well-being by monitoring their health, administering medications, and overall improving welfare and care [17,18].

In the context of farm animals, smart technologies play a crucial role in ensuring safety and welfare. Sensors, artificial intelligence, virtual fencing, robotic milking, and automatic feeding collectively contribute to improving farm animal safety and welfare. Additionally, smart farm technologies not only increase productivity but also reduce environmental impact, fostering a positive public perception [19,20].

As we look ahead, the role of smart technologies in animal care and health maintenance is expected to grow significantly [21–23]. These technologies will continue to advance, further improving animal health and welfare, and contributing to a more sustainable and compassionate approach to animal care.

Livestock feeding is undergoing a revolutionary transformation through smart feeding systems. This innovative technology, powered by sensors, data analysis, and automation, significantly enhances animal feeding precision. The benefits of this high-accuracy approach extend to improved animal health, increased productivity, and enhanced profitability [24,25].

Smart feeding systems leverage sensors and data analysis to elevate animal feeding programs, actively monitoring for any feeding issues and taking corrective actions as needed [26,27]. Notably, these systems contribute to heightened productivity while concurrently reducing labor costs. The integration of data analysis and machine learning plays a pivotal role in refining feed composition, ultimately boosting feed income and mitigating adverse environmental impacts [28,29].

The adoption of needs-based feeding is a key strategy in optimizing animal health, productivity, and environmental impact. Probes and sensors closely monitor feed and water consumption, ensuring a well-maintained nutritional balance. Additionally, feed management, coupled with graphical analysis, improves feed efficiency by optimizing composition, schedules, and water quality, ultimately contributing to heightened resource efficiency [30,31].

The implementation of smart feeding systems, characterized by the utilization of sensors, data analysis, and automation, significantly enhances livestock productivity and profitability.

The field of smart animal monitoring technologies is experiencing significant growth. These technologies gather real-time data to monitor and enhance both animal productivity and the overall environment. Revolutionary sensing and tracking technologies play a key role in animal monitoring, with image analysis and remote sensing allowing for the assessment and tracking of animal conditions on farms without breeder intervention. This, in turn, aids in evaluating animal welfare [32,33].

The integration of intelligent sensing and analysis technologies holds great promise for transforming animal welfare. This has positive implications for both animal farm management and conservation initiatives. The rapid advancement of bio-sensing and analysis technologies is giving rise to innovative applications that can contribute to the improvement of animal production. These applications cover a spectrum of enhancements in animal health and welfare [34–36]. In essence, smart animal monitoring technologies are driving a revolution in animal welfare, offering benefits to both animal farm management and conservation efforts.

Technological advancements in smart animal management and reproduction aim to enhance animal productivity, well-being, and promote sustainable farming practices. Sensors and data analytics play a crucial role in refining animal monitoring, breeding, and genomic selection. The integration of robotics and smart technology contributes to increased farm efficiency, productivity, and improved animal welfare [37,38]. Moreover, artificial insemination and selective breeding are employed to enhance animal production,

quality, and welfare. The implementation of smart chips and electronic tracking systems further refines animal monitoring and herd management [39,40].

Anticipated growth in animal identification systems within livestock production is expected to amplify productivity and quality. It is essential to consider the ethical and privacy implications associated with animal identification systems to ensure ethical treatment and compliance with local laws [41,42]. Overall, smart animal management and reproduction technologies utilize sensors, data analytics, robotics, and other technologies to advance animal productivity, well-being, and sustainable farming practices.

Smart farming applications play a crucial role in improving livestock marketing and distribution. They achieve this through various features such as tracking, data analysis, remote monitoring, and effective marketing and branding strategies. These applications offer real-time information to both farmers and retailers, ensuring the safety and traceability of produce.

By facilitating instant access to relevant data, smart farming applications empower farmers and retailers to make informed decisions. These decisions extend to crop planning, livestock management, and overall logistical operations. The result is a significant enhancement in efficiency within the animal production industry, ultimately leading to increased sales and profits [43–45].

To effectively market smart animal products, it is essential to highlight the benefits for both the animals and their owners, making care more convenient. Employing paid advertisements that focus on product benefits and visuals can be an impactful strategy. Additionally, creating high-quality content that educates and attracts customers is crucial. Staying abreast of industry trends ensures that marketing efforts remain relevant and effective.

Successful marketing of smart animal products plays a pivotal role in raising awareness of their quality [46–48]. In summary, smart farming applications contribute to livestock marketing and distribution improvements by providing real-time information and facilitating enhanced decision-making processes.

Smart animal production farms use technology to advance agriculture economically and socially. Smart farms use the internet of things (IoT), data analytics, and automation to improve operations and animal well-being, saving money and boosting efficiency. Smart farms also improve animal welfare, labor conditions, and sustainability, using technology and data to create a more sustainable and socially responsible future for agriculture [49–51]. Smart animal production farms use technology to improve agriculture economically, socially, and environmentally.

### 3.2. Practical Examples of Smart Animal Production

Smart healthcare solutions for animals, such as remote monitoring and treatments, can improve timely care and reduce the need for physical veterinary visits, making the process more sustainable and efficient [52]. Smart healthcare solutions for animals are in their early stages of development but have the potential to revolutionize animal care by providing the early detection of health problems. Smart healthcare solutions for animals are still in their early stages of development, they have the potential to make a significant positive impact on the lives of pets and their owners.

Utilizing facial recognition and machine vision technologies for observing animal behavior and well-being is crucial for ensuring animal welfare. This approach aids in identifying nutritional requirements and detecting potential productivity issues [53,54]. It is essential to emphasize that these technologies are not intended to substitute human interaction with animals. Humans should remain actively engaged in the care and monitoring of all animals, including those being observed through these technological means.

Radio-frequency identification (RFID) and the Global Positioning System (GPS) play a crucial role in livestock tracking, enhancing traceability, improving food safety, and supporting sustainable practices by facilitating precise resource allocation [55,56]. The utilization of RFID and GPS livestock tracking represents positive advancements for the

industry, with the potential to enhance efficiency, promote humane practices, and contribute to overall sustainability.

Milking machines, when used to milk cows at a set time each day, contribute to enhanced efficiency, milk quality, and sustainability, leading to energy savings [57]. These machines serve as valuable tools for dairy farmers, positively impacting efficiency, milk quality, and sustainability.

Smart livestock collars monitor activity levels, body temperature, and signs of illness or distress for animal well-being [58]. Smart livestock collars have the potential to play a significant role in improving animal well-being. By providing farmers and ranchers with real-time data about the health and behavior of their animals, smart livestock collars can help to identify and address potential problems early on, before they become more serious.

Farmers track animal veterinary history, growth rates, and performance metrics to make data-driven decisions for improved animal health and productivity [59]. Tracking and analyzing data can help farmers make better decisions about animal care, leading to healthier and more productive animals, reduced costs, and a more sustainable agricultural system.

The use of data analytics and machine learning algorithms aids farmers in minimizing resource wastage, optimizing breeding practices, and promoting sustainable animal husbandry by predicting diseases and devising effective breeding strategies [60]. This application of data analytics and machine learning has the potential to revolutionize agriculture, fostering a more sustainable, ethical, and profitable environment for farmers.

Smart fencing solutions can control animal movement remotely or independently, enhancing animal management and promoting resource conservation [61,62]. Smart fencing solutions have the potential to revolutionize animal farming, making it more sustainable and profitable.

### 3.3. Developments in Smart Animal Agriculture

Advancements in ensuring the health and safety of animals have made significant strides, incorporating innovations such as wearable sensors, AI-driven disease diagnosis, predictive analytics, IoT applications for livestock, virtual reality training for veterinarians, satellite tracking, wildlife drones, biometrics for tracking, robotic surgery, and gene editing [63,64]. These remarkable progressions in animal healthcare seamlessly blend technology with compassion, offering a promising outlook for the well-being and conservation of animals, spanning from sensor technologies to gene editing.

Smart feeding systems use sensors, data analysis, and automation to improve animal feeding, benefiting feed efficiency, sustainability, and animal welfare. They employ RFID tagging, sensor analysis, automation, robotic feeders, and precision feeding. Future advances include AI-driven decisions, personalized feeding, and blockchain-based feed tracking [65,66]. Smart feeding systems are revolutionizing animal nutrition by optimizing efficiency and welfare through data-driven automation. The potential integration of AI and blockchain promises even more personalized and sustainable approaches in the future.

Revolutionary smart monitoring and control techniques for animals are reshaping the landscape of animal care. This integration of sensors, AI, robotics, and precision agriculture is not only enhancing welfare and boosting productivity but is also fostering sustainability. The result is a transformative impact on both animal agriculture and animal ownership. This amalgamation of smart technology with animal care is a game-changer, paving the way for a future where animals thrive, and their caregivers excel [67,68]. These advancements hold the key to improved welfare and productivity, from agriculture to animal ownership.

Advanced animal management and breeding techniques are transforming the industry through the incorporation of sensors, robotics, AI, and genomics. These technologies aim to boost animal productivity, welfare, and sustainability by offering real-time data, automating tasks, analyzing patterns, identifying genes, and optimizing breeding practices. This transformation of animal husbandry is heralding a brighter future [69]. The fusion of technology with animal management is redefining the industry, ensuring a more sustainable

and welfare-focused future. From instantaneous data to genetic insights, these innovations are paving the way for enhanced animal husbandry practices.

The advent of smart animal production technology has ignited a revolution in product marketing and distribution, enhancing efficiency, traceability, and consumer awareness. Real-time data enables precise feeding and data-driven decision-making. Brand visibility and customer engagement are amplified through social media, direct sales, and partnerships. Crucial pillars in this transformation include sustainability and regulatory compliance. The industry is poised for further advancement with the promise of AI, ML, and VR [70,71]. In essence, smart animal production technology is reshaping the landscape of product marketing and distribution, elevating efficiency, traceability, and consumer engagement. The industry's ongoing transformation is anticipated to reach new heights with the integration of AI, ML, and VR, all the while emphasizing sustainability and compliance.

Precision livestock farming (PLF) utilizes advanced technology to enhance animal welfare, increase productivity, and foster sustainability in agriculture. This approach yields economic and societal benefits. Precision livestock farming incorporates various tools such as individual animal monitoring, precise feeding systems, disease detection algorithms, robotic milking, and automated feed distribution. These tools collectively reshape animal farming, resulting in greater efficiency and consideration for animal well-being [72–74]. The transformative impact of precision livestock farming is evident in its ability to revolutionize agriculture by seamlessly integrating technology with animal welfare and sustainability. This ranges from individual monitoring to the implementation of robotic milking, ultimately molding the future of farming towards increased efficiency and humanity.

### 3.4. Technological Devices in Animal Production

Contemporary livestock management seeks to boost productivity, optimize operations, and promote animal welfare. In the domain of animal management, diverse smart technologies cater to specific needs, as detailed in Table 1. These technologies can be broadly classified into distinct categories.

The integration of smart agriculture and animal management technologies is transforming the way we manage livestock. As technology continues to evolve, we can expect even more innovative solutions that will further revolutionize the animal management industry and contribute to a more sustainable and efficient agricultural sector.

The use of technological devices in livestock farming offers numerous benefits. First, it contributes to improving productivity by enhancing production efficiency and maximizing growth rates. Secondly, these devices enhance efficiency by reducing the need for labor and increasing production levels. Technological devices also play a vital role in improving animal care by ensuring the provision of proper nutrition and necessary health care [93,94].

In terms of increasing productivity, these technologies contribute to enhancing livestock production by improving feed quality and providing superior health care. Additionally, technological devices play a role in improving efficiency by reducing manual work and making better use of resources. Finally, these technologies enhance animal welfare by creating an improved environment and providing improved health care, contributing to the overall quality of life of livestock [95,96].

The adoption of technological devices in livestock farming presents various challenges. Firstly, the high cost of these devices can be a significant obstacle for some breeders. Moreover, the need for frequent maintenance adds to the already demanding workload of breeders. Another issue is the reliance on energy, especially in areas with an unreliable electricity supply. Additionally, the vulnerability of these devices to breakage or malfunctions raises concerns about their reliability, potentially affecting the overall productivity of livestock farming operations [97–99].

In the realm of livestock farming, technological devices present a plethora of advantages, outweighing any potential drawbacks by improving productivity, efficiency, and animal welfare. While these devices bring about significant benefits, it is imperative to

thoughtfully evaluate factors such as costs, maintenance, and energy dependence before integrating them into farming practices.

**Table 1.** A comprehensive overview of animal technology devices in animal care.

| Category | Technology | Description |
|---|---|---|
| Health Monitoring | Health Wearables | This device continuously monitors and improves animal well-being by gathering and interpreting a range of health-related data. These tools employ technology to provide valuable insights into an animal's physical condition, behavior, and overall well-being. In addition, they are used to track animal health parameters such as heart rate and activity levels, enabling the early detection of potential health problems [75]. |
| | Real-time Monitoring Devices | State-of-the-art animal monitoring devices continuously observe and analyze animal welfare data, providing real-time insights into their health and well-being. It delivers instant data on animal health, enabling timely interventions and improved care [76]. |
| | Livestock Sensors | Livestock sensor devices are used to monitor and manage the health of animals in agricultural settings. They collect data on animal health and behavior, which can be used to make informed decisions about animal care [77]. |
| | Advantages | Key aspects include real-time data tracking, early detection of diseases, analysis of vital signs, and proactive health management for both livestock and companion animals. |
| | Disadvantages | Challenges include limited accuracy in early disease detection, potential errors in data interpretation, issues with device accessibility, and difficulties in the continuous monitoring of free-ranging animals. |
| Feeding Management | Automated Feeding Systems | Automated feeding systems (AFS) are innovative devices that streamline and enhance animal feeding using automation technology. AFS provides precise and scheduled feedings, ensuring optimal nutrition and resource management. They play a crucial role in modern animal husbandry by promoting consistency and improving efficiency [78]. |
| | Smart Feeders | Smart feeders are cutting-edge devices that revolutionize animal feeding using advanced technology. These innovative feeders provide intelligent and automated feeding solutions, offering convenience, precision, and remote control. They enhance animal care and well-being while optimizing nutrition [79]. |
| | Advantages | Key elements include precise feeding, strategic nutritional planning, quality-controlled feeds, and automated feeding systems to enhance the health and performance of animals. |
| | Disadvantages | Challenges may include imprecise feeding, nutritional imbalances, feed quality variations, and automated system issues. |
| Welfare Monitoring | Behavior Sensors | Animal behavior sensor devices are sophisticated tools that use advanced sensor technology to observe and analyze animal behavior. These devices provide real-time insights into animal behavior, helping researchers, caretakers, and anyone involved in animal management better understand their well-being [80]. |
| | Robotic Herding | Robotic herding devices represent a technological leap in livestock management, autonomously guiding animals using sensors and intelligent algorithms. They streamline herding, minimize stress, and optimize overall herd management, combining the precision of robotics with practical herding for a modern approach to animal husbandry [81]. |
| | Advantages | Continuous monitoring involves real-time data collection, behavior analysis, and tracking of health indicators. |
| | Disadvantages | Necessitate constrained precision, risk of sensor-induced stress, and difficulties in interpreting behavioral data. |
| Decision Support | AI Equipment | AI-equipped animal devices integrate artificial intelligence into animal care, employing advanced algorithms to analyze health, behavior, and overall well-being data. These devices provide valuable insights and personalized solutions for individual animals or groups, contributing to enhanced decision-making and proactive health monitoring in animal care, improving overall efficiency in animal management [82]. |
| | Advantages | Provide data-driven insights, predictive analytics, personalized recommendations, and optimizes health. |
| | Disadvantages | Necessitate restricted data precision, susceptibility to algorithmic biases, and reliance on accurate input data. |
| Remote Monitoring | Remote Monitoring Systems | Animal remote monitoring systems employ remote sensing technologies to comprehensively track and analyze various aspects of animal health, behavior, and environmental conditions. Equipped with sensors and communication capabilities, these devices enable real-time monitoring and data transmission from a distance. Whether used in wildlife conservation or animal care, they provide researchers and caretakers with valuable, timely information for proactive management and swift responses to changes in an animal's conditions [83]. |
| | Advantages | Provide real-time tracking, environmental sensing, health parameter monitoring, and automated alerts. |
| | Disadvantages | Require addressing connectivity issues, mitigating data security concerns, ensuring device reliability, and minimizing the potential for false alarms. |
| Security | Remote Cameras | Animal remote camera systems are advanced devices for observing animal activities. With high-quality cameras, motion detection, and night vision, these systems provide continuous monitoring in various environments. Transmitting live or recorded footage remotely, they offer insights into animal behavior, habitat use, and interactions. Whether in wildlife research, livestock management, or animal observation, these systems provide an efficient and non-intrusive way to study animals in their natural or domestic settings [84]. |
| | Advantages | Provide high-resolution imaging, infrared night vision, motion detection, and capabilities for remote viewing. |
| | Disadvantages | Require addressing limited battery life, weather vulnerability, potential for signal interference, and upfront cost. |

**Table 1.** *Cont.*

| Category | Technology | Description |
|---|---|---|
| Inventory Management | RFID-tagged Systems | Animal RFID-tagged systems use specialized devices for animal identification and tracking through radio-frequency identification (RFID) technology. With RFID tags containing unique codes attached to animals, RFID readers wirelessly collect and process information for efficient and accurate identification. Widely used in livestock management, wildlife research, and animal tracking, these systems streamline data collection, enhance security, and contribute to the overall management and well-being of animals [85]. |
| | Livestock Trackers | Animal livestock trackers are specialized devices that utilize GPS or other tracking technologies to provide real-time location information for individual animals within a herd. With features like geofencing and activity monitoring, these trackers enhance overall herd management, prevent loss, and optimize grazing patterns, ensuring efficient livestock monitoring in various environments [86]. |
| | GPS Management | Animal GPS management devices use GPS technology for monitoring and controlling animal movements. Equipped with GPS modules, they enable real-time tracking and location identification of individual animals. With applications in wildlife research, and animal management, these devices contribute to enhanced security, efficient herd management, and the overall well-being of animals by providing valuable insights into their locations and movements [87]. |
| | Advantages | Provide automated record keeping, RFID tracking, real-time data updates, and analytics for inventory management. |
| | Disadvantages | Require addressing the potential for RFID malfunctions, minimizing data entry errors, simplifying system complexity, and managing initial setup costs. |
| Farm Management | Livestock Management Software | Livestock management software devices generally simplify and improve livestock care through advanced applications for health monitoring, breeding records, and feeding schedules. Featuring user-friendly interfaces, these devices enable efficient data entry, analysis, and decision-making for ranchers. By centralizing information and automating tasks, they enhance productivity, resource utilization, and the overall well-being of the livestock. Whether on small-scale farms or large agricultural operations, these devices are crucial for modern livestock management [88]. |
| | Advantages | Provide livestock health tracking, feed optimization, breeding management, and analytics for production. |
| | Disadvantages | Require addressing data security risks, reducing equipment dependency, and mitigating the potential for inaccurate data entry. |
| Environmental Control | Environmental Controls | Animal environmental controls devices optimize environmental conditions for animals, regulating factors like temperature, humidity, ventilation, and lighting within enclosures. By ensuring a comfortable environment, these devices contribute to the well-being, health, and productivity of animals, playing a crucial role in creating optimal living conditions in places such as livestock barns, poultry houses, or animal habitats [89]. |
| | Advantages | Ensure animals' well-being through temperature control, humidity management, ventilation, and optimal lighting for comfort. |
| | Disadvantages | Limited precision in environmental settings, adaptability issues across various species, potential energy inefficiency, and difficulty accommodating individual animal preferences. |
| Reproductive Management | Hormone Monitors | Animal hormone monitor devices track and analyze hormonal levels in animals, utilizing advanced technology to measure concentrations of various hormones. They provide insights into reproductive cycles, stress levels, and overall health, aiding in fertility management, breeding programs, and overall well-being. Whether in wildlife research, or veterinary care, these devices play a pivotal role in optimizing hormonal balance for improved animal health and reproduction [90]. |
| | Advantages | Key aspects include synchronizing estrus, implementing artificial insemination, optimizing breeding programs, and monitoring reproductive health. |
| | Disadvantages | Challenges encompass fluctuating success rates in artificial insemination, precision concerns in estrus synchronization, limited adaptability across species, and potential health risks linked to intensive reproductive interventions. |
| Supply Chain Management | Smart Distribution Systems | Animal smart distribution systems devices use smart technology to optimize resource distribution, automating dispensing based on schedules or conditions for efficient delivery. Whether in zoos or animal care, they provide a technologically advanced solution for animal resource management. They also optimize the flow of animal products from farm to consumer, reducing costs and improving efficiency [91]. |
| | Advantages | Encompassing inventory tracking, logistics optimization, quality control, and streamlined distribution for efficient sourcing of animal-related products. |
| | Disadvantages | Challenges encompass supply chain disruptions, insufficient temperature control during transportation, quality assurance issues, and obstacles in real-time inventory tracking. |
| Packaging | Automated Packaging Systems | Animal automated packaging systems devices use advanced technology to automate packaging, incorporating features like automated weighing, sorting, and sealing for accuracy and speed. Whether used in animal food packaging or other animal-related goods, these devices enhance productivity, reduce labor costs, and improve overall efficiency in the packaging phase of animal product manufacturing. They streamline the process, saving time and improving efficiency [92]. |
| | Advantages | Key features of animal product packaging include secure, durable, and hygienic designs to uphold product integrity, preserve freshness, and comply with safety standards. |
| | Disadvantages | Packaging limitations may involve environmental concerns, inadequate protection against contamination, limited recyclability, and challenges in balancing cost-effectiveness with sustainability. |

*3.5. Revolutionizing Animal Farming*

Smart devices for animal health encompass a variety of tools such as activity trackers, GPS collars, remote monitoring systems, smart feeders, health wearables, pet-access doors,

tank regulators, telemedicine services, livestock sensors, and health apps. Collectively, these devices work to guarantee the safety and well-being of animals [100,101].

In the realm of animal farming, smart feeding technology comprises automated feeders, consumption sensors, nutritional analyzers, RFID-tagged systems, precision equipment, monitoring software, custom mixers, and real-time monitoring. These technologies streamline the feeding process, ensuring precise nutrition and efficient feed management for animals [102,103].

Smart animal monitoring and control involve a variety of devices, including tracking collars for location, wearables, remote cameras/sensors, environmental controls, gates/fences for access, automated feeding systems, livestock trackers, behavior sensors, robotic herding, and cloud-based management. Collectively, these technologies enhance animal welfare and streamline care [104,105].

In the realm of intelligent animal breeding and management, a convergence of AI equipment, genetic testing tools, embryo transfer devices, hormone monitors, smart identification systems, health monitors, automated feeding, data analytics platforms, and livestock management software takes place. This integration aims to optimize breeding practices and improve overall animal care. Collectively, these technologies work together to ensure the production of healthier offspring and the efficient management of animal populations [106–108].

The smart distribution of animal farm products integrates diverse technologies, such as RFID/barcodes and smart packaging sensors for tracking and quality control during transit, coupled with GPS management and supply chain software for efficient delivery. IoT sensors monitor product quality, while blockchain ensures transparent tracking, complemented by automated packaging and cloud-based inventory for real-time tracking. Together, these technologies ensure reliable, high-quality distribution throughout the supply chain [109–111].

The socio-economic impact of smart animal farming revolves around essential technologies, encompassing data analytics and remote monitoring for informed decision-making and animal health. This includes automated feeding, health wearables, management software, genetic testing, and the utilization of supply chain and blockchain technologies for quality enhancement and transparent tracking. Collectively, these technological advancements serve to boost productivity, enhance animal welfare, and improve resource management, thereby fostering positive economic and social outcomes within the farming sector [112,113].

Smart technologies revolutionize animal farming, transforming health, feeding, monitoring, breeding, and distribution. These innovations enhance animal welfare, optimize resource management, boost productivity, and improve economic outcomes, promising a more sustainable, efficient, and humane agricultural future.

*3.6. Advantages and Disadvantages of Smart Animal Production*

Smart healthcare for animals holds promise with its ability to detect diseases early, enhance nutrition, provide real-time tracking, and improve overall welfare. However, obstacles such as high costs, privacy concerns, data accuracy, user errors, and social acceptance must be addressed [114,115]. While there is exciting potential for advancements in animal health, it is crucial to focus on addressing challenges such as costs, privacy, and social acceptance to ensure widespread success.

Smart feeding systems present a gamut of technological advantages poised to enhance efficiency, productivity, and animal welfare in production farms. Yet, it is crucial to meticulously weigh the potential downsides—think initial investment costs, maintenance demands, and the ever-looming specter of data security concerns—before taking the plunge into adopting these systems [116,117]. Smart feeding systems offer farm benefits, but weighing initial costs, maintenance, and data security is key for informed adoption decisions.

Implementing smart techniques for animal monitoring and control can yield technological advantages that enhance health, welfare, and productivity. Nevertheless, it is

essential to weigh the drawbacks, such as initial expenses, technical expertise, data security concerns, potential over-reliance, and the acceptance of these technologies by animals. Achieving success in this endeavor depends on meticulous planning, effective implementation, and thorough training to optimize benefits and mitigate potential drawbacks [118,119]. While smart animal monitoring contributes to well-being and productivity, successful implementation requires thoughtful consideration of costs, security, and the adaptability of animals.

Smart animal management and breeding techniques offer a tech-savvy edge, enhancing efficiency and outcomes in farming. Potential drawbacks like upfront costs, technical challenges, data security risks, and the need for animal adaptation may become hurdles. It is a balancing act between reaping the benefits and managing the potential downsides [120,121]. Smart techniques elevate animal management, but navigating upfront costs, technical challenges, and ensuring animal adaptation is crucial for balanced success.

Smart animal tech offers benefits such as traceability, supply chain, marketing, welfare, and compliance, and disadvantages such as upfront costs, privacy issues, need for tech expertise, errors, and public perception [122,123]. Smart animal tech brings valuable benefits, but upfront costs, privacy concerns, and public perception should be considered carefully for a well-rounded approach.

Smart animal farming presents a promising strategy to boost agricultural productivity, enhance animal welfare, and support environmental sustainability. Nevertheless, a thoughtful examination of potential drawbacks, such as upfront investment costs, data management challenges, animal welfare concerns, and societal implications, is crucial for its successful and responsible implementation [124–126]. While smart farming holds significant potential for productivity and sustainability, addressing costs, data challenges, and ethical considerations is vital for its responsible implementation.

### 3.7. Technology Alternatives in Smart Animal Production

In smart animal production, alternatives to technology include traditional methods like animal husbandry practices, manual record-keeping, and visual inspections. Practices such as natural grazing management and manual feeding allow farmers to manage livestock without electronic monitoring. Traditional expertise is highlighted through selective breeding and non-digital health monitoring. While these approaches may lack high-tech efficiency, they emphasize the integration of time-tested methods with modern technologies for comprehensive livestock management [127].

Smart animal production farms, with their technological advancements, offer significant benefits. However, ethical and sustainable food production can also be achieved through alternative methods. Enhancing animal welfare, minimizing environmental impact, and fostering community connections are attainable by incorporating genetic selection strategies, implementing effective management practices, exploring alternative feedstuffs, prioritizing local sourcing, and engaging in direct marketing [128–130].

While smart farms bring technological benefits, ethical and sustainable food production requires a diverse set of approaches, encompassing genetics, management practices, alternative feedstuffs, and local sourcing for improved welfare and reduced environmental impact.

### 3.8. Livestock Accuracy Enhancing Applications

Smart applications have the potential to enhance animal welfare in agriculture through various functionalities. These include monitoring the health and well-being of animals, implementing precision feeding and watering systems, overseeing environmental conditions, automating tasks, and collecting and analyzing data. Notable examples of such applications encompass smart collars, robotic milking systems, precision feeding systems, and environmental monitoring systems [131].

In the realm of livestock management, smart applications play a crucial role by tracking individual animal health, providing personalized feeding and breeding guidance,

conducting behavior analysis for early issue detection, managing inventory, regulating environmental conditions, and offering market insights. These tools contribute to the efficient management of livestock, ensuring their health and productivity, and facilitating informed decision-making [132,133].

Smart applications bring a technological boost to animal welfare in farming, utilizing tools such as smart collars and robotic systems to deliver precise care and data-driven insights.

Precision livestock farming (PLF) utilizes data to enhance animal health, welfare, and productivity in large-scale and intensive systems. While PLF holds great potential, challenges like cost, data management, integration, and animal welfare persist. Nevertheless, progressively PLF is employed for automated milking, precision feeding, environmental monitoring, and disease detection. Overall, PLF stands as a promising technology with the potential to revolutionize livestock management [134–136]. Precision livestock farming has transformative potential in enhancing animal health and productivity, despite challenges like cost and welfare concerns. Its applications in automated milking, precision feeding, and disease detection mark it as a promising technology for revolutionizing livestock management.

As technology continues to develop, we can expect to see even more innovative livestock accuracy-enhancing applications that will provide even more benefits to farmers, ranchers, and other stakeholders.

Smart technologies for healthcare and safety in the animal domain are advancing swiftly, providing various advantages such as enhanced animal welfare, lowered veterinary expenses, increased productivity, and improved safety. These technologies find application in remote monitoring, precision medicine, behavioral monitoring, and safety monitoring. Noteworthy examples include PetPace, a novel pet-related device or service, Tractive, a GPS animal tracking device, and Cowlar, a smart collar tailored for cows [137,138].

Smart feeding systems (SFSs) employ sensors, artificial intelligence (AI), and machine learning (ML) to automate and enhance livestock feeding processes. These systems have the potential to enhance feed efficiency, lower feed costs, promote animal health and welfare, decrease labor expenses, and elevate the quality of meat and milk. Additionally, SFSs can contribute to the reduction of the environmental footprint associated with animal production and enhance traceability. With the increasing affordability and sophistication of SFSs, a broader adoption in the animal production industry is anticipated [139].

Advanced techniques for the monitoring and controlling of animals are on the rise, employing sensors, actuators, and AI to collect data on behavior, health, and the environment. These methods provide real-time insights into health, behavior, environmental conditions, predator protection, and location tracking. The advantages include improved animal welfare, decreased caretaker workload, and heightened efficiency. However, there are existing challenges such as cost, data privacy, and stakeholder acceptance. Despite these obstacles, these technologies have the potential to bring about significant positive impacts on animals [140,141].

Utilizing advanced technology in animal care enhances welfare, productivity, and sustainability by incorporating AI, data analytics, and sensors. Precision feeding, health monitoring, and managing reproductive and behavioral aspects contribute to the well-being of animals, resulting in increased yields, cost efficiencies, and environmental friendliness. Despite challenges related to costs and standardization, there is the potential to revolutionize animal agriculture, paving the way for a more humane and sustainable future for food. Leading the charge in this technological transformation are key players such as Allflex, Daftras, Delphi Animal Health, Elysis, and Zoetis [142,143].

Smart animal production farms use technology to automate and optimize livestock production processes. This can lead to more sustainable, healthier, and more nutritious animal products. Marketing and distributing smart animal products can be challenging, but producers can overcome these challenges by focusing on the benefits of these products to consumers and educating them about smart farming technologies. Smart farming

technologies can be used to market and distribute smart animal products through real-time data monitoring, traceability, e-commerce, social media, content marketing, and partnerships [144].

Smart animal farming is a rapidly evolving field poised to revolutionize the livestock industry by boosting efficiency, reducing costs, improving profitability, enhancing animal welfare, mitigating environmental impact, and ensuring food security. As these technologies become more affordable, they are likely to be widely adopted, fostering a more sustainable, efficient, and humane livestock industry globally [145–147].

The use of smart technological applications in animal production has the potential to revolutionize the industry, leading to increased productivity, improved animal welfare, enhanced environmental sustainability, and greater profitability for farmers.

### 3.9. Blockchain in Animal Production Farms

Blockchain has the potential to revolutionize the livestock sector, enhancing its efficiency, transparency, and sustainability, providing numerous opportunities and benefits for farmers, consumers, and other stakeholders. Nevertheless, it is crucial to acknowledge the challenges associated with this technology, including high initial investment costs, cybersecurity risks, a shortage of skilled labor, and regulatory hurdles [148]. Investing in research and development is essential to tackle these challenges, and collaboration is key to establishing a supportive environment for the widespread adoption of Blockchain technology. As technology continues to develop and mature, these challenges will likely be addressed, paving the way for Blockchain to play a transformative role in the livestock sector. The potential advantages of Blockchain technology in the live-stock sector far surpass the challenges [149,150].

In the future, we anticipate expanded uses of blockchain in animal production, blockchain enhances traceability, prevents disease, and streamlines supply chains, benefiting both producers and consumers. It fosters sustainability, aids disease prevention, and supports financial inclusion for farmers in developing nations. This technology has the potential to significantly improve the efficiency and transparency of the livestock industry, with ongoing innovations expected.

### 3.10. Opportunities, Challenges, Solutions, and Benefits for Smart Animal Production Farms

3.10.1. Opportunities

The emergence of smart animal production farms is on the rise, bringing the promise of heightened efficiency, increased yields, and minimized waste [151,152]. These farms not only enhance animal welfare through data-driven monitoring and management but also contribute to environmental conservation by adopting sustainable farming practices [153,154]. The integration of technology into agriculture not only creates job opportunities in tech development, maintenance, and farm management, thereby bolstering local economies but also enables informed decision-making and optimal resource allocation through the use of big data from smart farms [155–157].

Automation and technology play a pivotal role in improving farm labor conditions, while the implementation of smart farming practices can enhance quality and traceability, meeting the demands of safety-conscious consumers. Furthermore, smart farms act as catalysts for research and innovation, bridging the gap between agriculture and technology [158–160]. By emphasizing their environmentally friendly and socially responsible practices, farms can establish themselves as sustainable and responsible brands, appealing to conscious consumers.

Governments and regulatory bodies may incentivize and provide support for smart farming initiatives, thereby propelling growth and influencing the future of agriculture [161,162]. Smart livestock production farms present opportunities to enhance the efficiency, sustainability, and ethical standards of animal agriculture, addressing key challenges in 21st-century agriculture. Ultimately, these opportunities empower farmers to improve animal welfare, increase productivity, and mitigate environmental impact.

### 3.10.2. Challenges

While smart agriculture techniques hold immense potential to boost animal production, their widespread adoption faces several challenges. The high cost of advanced technologies can hinder their accessibility to smallholder farmers, who often operate on limited budgets. Data management and privacy concerns, as documented in various studies [163–165], pose additional barriers to smart agriculture adoption. Moreover, communication and infrastructure gaps in remote or underdeveloped areas further impede the implementation of technology-driven agricultural practices.

Education and training for farmers are crucial for successful smart agriculture integration, but resource scarcity and low literacy rates in certain regions limit access to such programs [166,167]. Furthermore, integrating and matching smart farming tools with existing agricultural systems can be a complex and resource-intensive endeavor, requiring careful planning and coordination.

To ensure the long-term sustainability and ethical implementation of smart agriculture, environmental considerations must be prioritized [168,169]. This includes minimizing the environmental footprint of agricultural technologies and ensuring that animal welfare is not compromised in the pursuit of increased productivity.

Specific to smart livestock farming, regulatory, data, and integration challenges demand attention. The establishment of standardized regulations, clarification of data ownership rules, and the facilitation of smart practice adoption among farmers are essential steps towards overcoming these hurdles [170,171]. Additionally, adapting to environmental variability, particularly in the face of climate change, is critical for the success of smart livestock farming.

Effectively addressing the multifaceted challenges of smart agriculture is paramount to unlocking its full potential for enhanced animal production, sustainability, ethical practices, and equitable access [172–174]. By devising strategies to overcome these barriers, we can pave the way for a more productive, sustainable, and equitable agricultural future.

Establishing smart animal production farms requires a comprehensive and well-organized approach that takes into consideration economic, technological, educational, and ethical factors. Common challenges include costs, usability issues, and concerns related to data privacy and security. Successfully overcoming these obstacles is vital for the successful implementation of these farms, emphasizing the importance of addressing these diverse elements within a cohesive strategy.

Drone farming on livestock farms can incur costs ranging from hundreds to tens of thousands of dollars, contingent on the farm's size and available equipment. These expenses encompass drones, data processing equipment, support, maintenance, power, storage, backup, internet, and network infrastructure. Despite the initial investment, the adoption of drones and digital data processing can yield numerous advantages, including increased productivity and enhanced decision-making.

It is essential to evaluate the precise costs and potential benefits in alignment with the unique requirements and objectives of the farm [175–178]. The cost of implementing smart techniques on livestock farms varies, offering significant potential benefits. Farmers should carefully assess costs and benefits to align with their unique needs and goals when integrating smart techniques on livestock farms.

The lifespan of smart farming technologies on livestock production farms ranges from 3 to 7 years, indicating a relatively short lifespan. To extend their longevity, several measures can be taken, such as providing proper maintenance and support, shielding them from harsh environmental conditions, and upgrading them to the latest versions [179,180].

Smart technologies in animal production offer opportunities for operational efficiency, animal welfare, environmental impact, economics, employment, and decision-making, but there are also challenges related to cost, data, privacy, connectivity, infrastructure, and education. The benefits of smart technologies in animal production include increased profitability, sustainability, and animal welfare while reducing environmental damage (Figure 2).

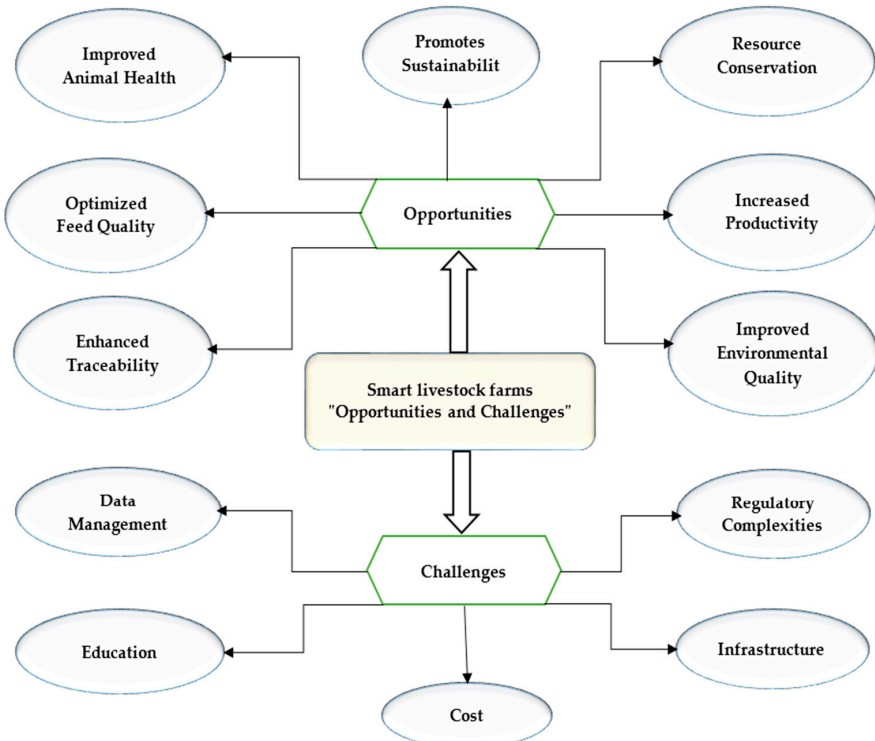

**Figure 2.** Flowchart of opportunities and challenges for smart livestock production farms.

### 3.10.3. Solutions and Benefits

The enhancement of livestock productivity can be achieved through the utilization of remote sensing, farm management systems, and technology. Smart nutrition and artificial intelligence also play vital roles, but the key factor is farmer training [181]. Moreover, the integration of smart technologies in animal production not only enhances profitability but also contributes to sustainability, animal welfare, and environmental considerations. However, it is imperative to carefully consider ethical aspects [182]. Solutions to challenges in smart agriculture for animal production involve making these technologies more affordable, offering training and support, and establishing standards for data privacy and security.

Smart livestock farms use technology to monitor and manage livestock remotely, providing farmers with a number of benefits, including improved animal health and welfare, increased productivity, reduced costs, improved sustainability, and increased profits [183]. In addition, smart livestock farms can help to improve food safety, increase transparency, and create jobs. Smart livestock farm applications offer a variety of solutions to improve the efficiency, sustainability, and welfare of livestock operations. By using these applications, farmers can make better decisions about their operations and produce more food with fewer resources.

Smart livestock technology can be beneficial for increasing farm efficiency and sustainability. Smart livestock farms use technology for solutions to remotely monitor, enhance animal welfare, increase productivity, improve cost efficiency, enhance sustainability, and increase profits. In addition, this technology enhances food safety, transparency, and employment opportunities (Figure 3).

While the adoption of smart agriculture technologies holds great promise for sustainable livestock production, it is essential to ensure that these technologies are accessible and affordable to a wide range of farmers, regardless of their scale of operation. Training and education are also key components to help farmers make the most of these innovations while promoting sustainable practices in the livestock industry [184–187]. The focus on smart agriculture for sustainable livestock, along with accessible tech and farmer education,

is key for inclusive and effective adoption. A holistic approach does not just address technology but also its integration and sustainability in farming practices.

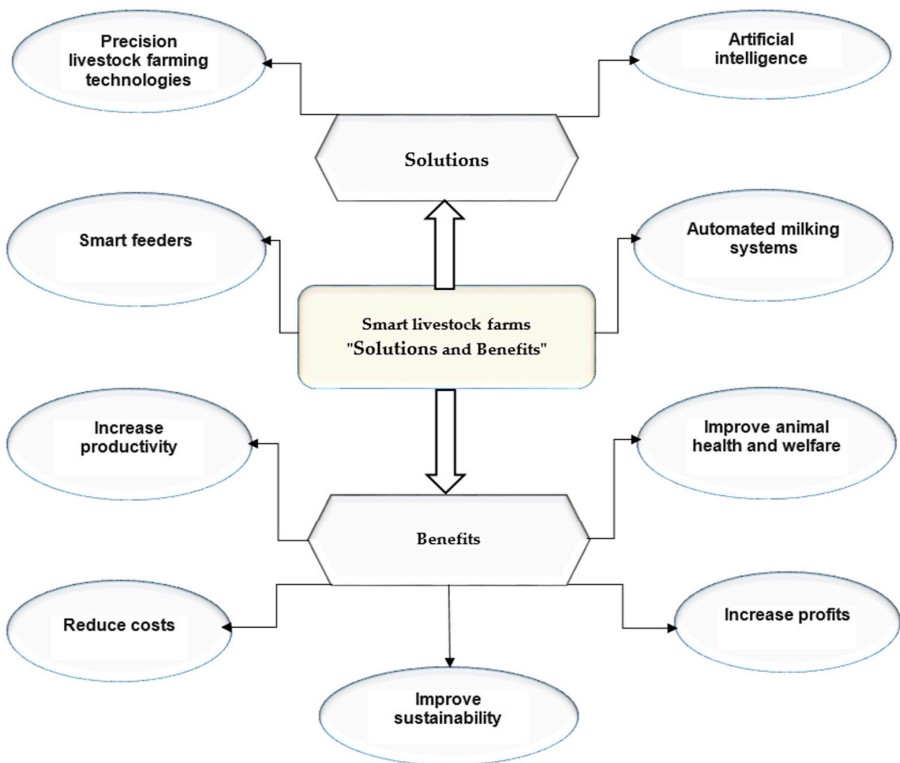

**Figure 3.** Flowchart of solutions and benefits for smart livestock production farms.

*3.11. Holistic Economic Approaches to Smart Livestock Farms*

Holistic economic strategies for intelligent livestock farming prioritize the integration of technological advancements and data-informed decision-making. The goal is to enhance productivity while simultaneously addressing environmental and social considerations. These methods emphasize resource efficiency, sustainability, and principles of the circular economy. They seek to diversify income through the creation of value-added products, leveraging data analytics for informed decisions, ensuring financial stability, and actively engaging with local communities.

By focusing on reducing environmental impact, staying responsive to market trends, and fostering a culture of continuous improvement, these approaches aim to create economically viable, environmentally responsible, and socially beneficial livestock farming systems. Recognizing the interconnectedness of these elements is crucial for the long-term sustainability and resilience of the agricultural sector [188–192].

In smart livestock farming, holistic strategies integrate technology and data to enhance productivity, promote sustainability, and diversify income streams. The overarching emphasis remains on achieving financial stability, fostering community engagement, and upholding environmental responsibility, all to ensure the long-term sustainability and resilience of agriculture.

Precision dairy monitoring technology on farms, influenced by IoT advancements, offers a means to choose cost-effective and strategic technology alternatives. An evaluation of existing livestock decision support systems (DSSs) reveals their focus on economic and environmental factors, with limited consideration for social aspects and governance. Additionally, there are unaddressed challenges related to IoT data management. Closing these gaps is essential for advancing economically sustainable livestock farming through DSS development [193–196]. Precision dairy technology and the IoT enable cost-effective

choices, yet current DSS systems overlook social and governance aspects. Addressing these gaps is crucial for the sustainability of livestock farming.

Agricultural industrialization is essential in advancing livestock production farms due to its capacity to drive efficiency and scale in operations, integrate cutting-edge technology for better management, ensure standardization and quality control, stimulate economic growth through job creation and market expansion, and, when managed responsibly, address environmental impact by implementing sustainable practices. This industrial shift not only meets the increasing demand for animal products but also contributes to global food security, making it a cornerstone in the evolution and optimization of livestock farming [197–200]. The agricultural industrialization in livestock production is multifaceted, encompassing efficiency, technology, quality, economic growth, environmental responsibility, and its role in global food security and the ongoing evolution of farming practices.

### 3.12. Future Research and Development Trends in Smart Livestock Farms

The trajectory of smart livestock farming is evolving through a persistent emphasis on innovation and the advancement of new technologies. These technologies empower farmers to improve animal welfare, boost productivity, and embrace sustainable practices. Essential areas for future research and development encompass precision livestock farming, artificial intelligence, robotics, and automation, along with sustainable practices [201,202]. Additionally, there is a need for continued research and development to seamlessly integrate diverse smart agriculture technologies. This integration will enable breeders to optimize their investments in smart farming technologies, fostering the creation of streamlined and efficient smart livestock farms while enhancing the sustainability of smart livestock production systems.

### 4. Conclusions

Smart agriculture is a burgeoning field with vast potential to elevate livestock production and contribute to sustainable development goals. The key advantages of this approach include enhancing animal welfare, improving resource efficiency, and increasing overall livestock production. However, addressing challenges related to cost, education, and policy is essential to ensure the success and widespread adoption of smart agriculture for sustainable livestock practices. By harnessing technology to collect, transmit, store, and analyze data, this approach has the power to revolutionize livestock management, providing real-time information, controlling conditions, and automating tasks. As these technologies continue to advance and gain broader acceptance, we can anticipate witnessing even more innovative and transformative applications in the years ahead, fostering a more efficient and sustainable future for livestock farming.

Smart agriculture technologies stand poised to revolutionize livestock production, ushering in a future marked by sustainability, efficiency, and profitability. While still in the nascent stages of development, these technologies encounter challenges such as cost and training. Despite these obstacles, the significant potential benefits indicate that smart agriculture technologies will play an increasingly vital role in the future of livestock farming. Offering a promising approach to addressing the challenges of sustainably feeding a growing population, smart livestock farming holds the key to transforming the sector. This research maintains an optimistic perspective on its capacity to revolutionize the livestock industry, contributing significantly to the enhancement of food security.

Smart animal production emerges as a promising solution to tackle the economic and environmental challenges inherent in traditional livestock production methods. Through the strategic application of technology, we have the capacity to optimize production efficiency, mitigate environmental impact, and secure the sustainability of livestock production for the benefit of future generations.

It is crucial to recognize that the economic and environmental outcomes of smart animal production are contingent upon factors such as the specific technologies adopted, the scale of operation, and the management practices employed. Thoughtful planning and

meticulous implementation are imperative to maximize the advantages and mitigate any potential drawbacks associated with this approach.

The findings of this research highlight that the integration of advanced smart agricultural practices, such as disease management, tracking, automated milking, and predictive analytics, has the potential to significantly improve the sustainability and growth of livestock.

## 5. Recommendations

Stakeholders in livestock production are advised to embrace the integration of smart agriculture with a positive mindset. The potential advantages of improving productivity and ensuring animal welfare through technological devices are significant. Nevertheless, it is essential to thoroughly assess the economic implications, considering both initial costs and ongoing maintenance expenses. Informed decision-making plays a crucial role in the successful integration of smart technologies into livestock operations. This research suggests that breeders should incorporate smart technology on their livestock farms, carefully considering all aspects, with a particular emphasis on economic factors. A comprehensive approach to adoption will contribute to optimizing the performance of animal production farms.

**Author Contributions:** M.D. introduced the research concept, outlined the general research plan, and managed review and editing tasks. S.S. and R.A.T. contributed to information and data collection, as well as data curation. A.N.A.-Y. and F.A.-B. drafted the initial research manuscript and performed validation. K.A.-N. authored, developed, and oversaw the publication and writing process, including the original draft preparation. All authors have read and agreed to the published version of the manuscript.

**Funding:** The research did not receive any funding.

**Institutional Review Board Statement:** This study does not require ethical approval.

**Informed Consent Statement:** Not applicable.

**Data Availability Statement:** Data are contained within the article.

**Acknowledgments:** The authors extend their sincere thanks to the University of Turku, Jerash University, Umm Al-Qura University, the National Agriculture Research Center, and the General Commission for Scientific Agricultural Research for their administrative support in completing this collaborative research.

**Conflicts of Interest:** The authors declare that they have no conflicts of interest.

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
