# Peer review of "Enhancing Animal Production through Smart Agriculture: Possibilities, Hurdles, Resolutions, and Advantages"

_ruminants, doi:10.3390/ruminants4010003_

Round 1

Reviewer 1 Report

Comments and Suggestions for Authors

The review presented is partly of good quality, and takes an interesting approach to exploring the advantages and opportunities offered by these technologies

However, it does not mention the real cost of these technologies, taking into account: on-board equipment, digital data processing equipment, the support needed to maintain these hard- and softwares and the induced energy costs, as well as the lifespan of the entire digital chain. In lines 280 to 283, a reservation is admittedly made, but it is insufficiently developed to modulate the partisan conclusion of this approach (as in line 346), which openly promotes a commitment to these technologies without considering all their aspects, particularly the economic ones.

I therefore recommend retaining the approach developed, but supplementing this literature review: either 1) with holistic studies including a serious economic approach, or 2) by modulating the discussion and the conclusion with information clearly pointing out the absence of serious holistic economic approaches and therefore the relativity and partiality of the approaches available.

Reviewer 2 Report

Comments and Suggestions for Authors

Manuscript Number: ruminants-2674424

Title of the Manuscript: Applications of Smart Agriculture to Improve Animal Production: Opportunities, Challenges, Solutions, and Benefits

Comments to authors

Line 7: edit “Collage”

Abstract: specify the methodology used for the review

 Introduction

·         A lot of repetitions (e.g., the sentence in lines 44-46 is a repetition of the one in lines 37-38. There are also redundancies in lines 49-68.

·         Missing:

o   the problem formulation is missing, and the objectives of the study need to be improved by being much more specific (what exactly did you do and for what purpose); .

o   Lines 70-71 “This study aims to improve sustainability and productivity in the animal production sector, benefiting both farmers and the broader agriculture industry’’ looks a development project’s objective

 Materials and Methods

·         Research questions should be placed in the Introduction section

·         it is not clear what methods are employed to review and synthesize relevant literature including the number of papers covered or synthesized

 Results and Discussion

The results should focus on exactly that you established in your review of literature, there was hardly any synthesis of findings related to the topic at all, that should be improved.

Comments on the Quality of English Language

The paper would benefit from some revisions and editing and polishing to address grammatical issues.

Reviewer 3 Report

Comments and Suggestions for Authors

Authors in their review article entitled " Applications of Smart Agriculture to Improve Animal Production: Opportunities, Challenges, Solutions, and Benefits" try to exploit the the use of precision livestock applications in favour of productivity, animal health and feeding. The text is very well structured, and divided into specific subsections that facilitated the reader. However, each of these sections in not well developed by authors that failed to described the most important advances that have been done in each of the mentioned fields. For example in section 3 they only referred to general statements for each of the subsection. Technological advantages should be described within each section mentioning various alternatives. They did not also make any comment about any possible disadvantages made from the implementation of smart agriculture. There is no link with welfare amelioration as well as with the early detection of i.e.  pathogenies or problems in a livestock unit, for which smart applications could assist farmers to decide at an early stage how to proceed. Further, there is no description about the possibilities of precision livestock application in extensive systems rather in intensive and what are the difficulties. The fact that the text is very well structured but with little information within  sections resembles to an AI software output,but this in not necessary a priori negative. A further enrichment of this could be done, highlighting any assist of such applications. In my opinion,  the thematic area of the review is very important but the text needs extensive revision.  Therefore, I recommend a rejection of the present status of the text, an extensive revision and re-submission.

Round 2

Reviewer 2 Report

Comments and Suggestions for Authors

I do not have any more comments.

Reviewer 3 Report

Comments and Suggestions for Authors

Authors have improved the text with new data however they stick to present findings using many section and subsections which does not facilitate the reader to follow and organize the information. Therefore i recommend a re-structure of the text especially for the sections 3.1-3.10. Authors should give all the proper information with in each technological advantage they describe (i.e. description, advantages, disadvantages, if it is used in livestock systems etc ).

Comments on the Quality of English Language

Authors should advise a native English speaker for better connection of the meaning between sections.
